# Health Behaviors in Austrian Apprentices and School Students during the COVID-19 Pandemic

**DOI:** 10.3390/ijerph19031049

**Published:** 2022-01-18

**Authors:** Teresa O’Rourke, Rachel Dale, Elke Humer, Thomas Probst, Paul Plener, Christoph Pieh

**Affiliations:** 1Department for Psychotherapy and Biopsychosocial Health, Danube University Krems, 3500 Krems, Austria; rachel.dale@donau-uni.ac.at (R.D.); elke.humer@donau-uni.ac.at (E.H.); thomas.probst@donau-uni.ac.at (T.P.); christoph.pieh@donau-uni.ac.at (C.P.); 2Department of Child and Adolescence Psychiatry, Medical University of Vienna, 1090 Vienna, Austria; paul.plener@meduniwien.ac.at; 3Department of Child and Adolescent Psychiatry and Psychotherapy, University of Ulm, 89075 Ulm, Germany

**Keywords:** COVID-19, health behaviors, adolescents

## Abstract

Background: The COVID-19 pandemic has disrupted our daily lives, which in turn has impacted health behaviors. Young people have been particularly affected. This study aimed to assess health behaviors in Austrian apprentices and high school students during the COVID-19 pandemic and whether vaccination willingness is affected by health behaviors. Methods: Two online surveys were conducted via REDCap with 1442 apprentices (female: 53.5%, male: 45.4%) from 29 March to 18 May 2021 and 563 school students (female: 79.6%, male: 18.6%) from 19 June to 2 July 2021. The two samples were matched to account for sociodemographic differences and analyses were run on the matched sample. Besides the health behaviors, namely, smoking, alcohol consumption, cannabis consumption, and exercise, health status and vaccination willingness were also assessed. Results: Health behaviors were affected by both education group and gender. Apprentices reported significantly more smoking than high school students and this difference was more pronounced in women (all *p* < 0.01). Alcohol consumption was higher in apprentices than school students, but only in women (*p* < 0.01). There was a trend for the two education groups to differ in their cannabis use as well (*p* = 0.05). Apprentices took part in more weekly exercise (*p* < 0.0001), but high school students reported better average health status (*p* < 0.001). When included in the same model, health behaviors did not affect vaccine willingness, but education group did, with high school students showing a higher willingness to receive the COVD-19 vaccine than apprentices. Conclusions: These findings support the argument that education type is an important factor for health behaviors, but this is also mediated by gender. Appropriate interventions for adolescents are needed to prevent adverse health behavior changes following the COVID-19 pandemic.

## 1. Introduction

The COVID-19 outbreak in December 2019 and the following restrictions have had detrimental effects on the lives of people all over the globe [1]. The pandemic has cost the lives of over 4.5 million people so far [2], and governments have implemented lockdown measures and social distancing policies to slow the further spread of the disease. In Austria, a first lockdown was implemented from 16 March 2020 to 30 April 2020, which was followed by further strict lockdown measures from 3 November 2020 to 8 February 2021, due to rising COVID-19 cases.

These measures have rapidly affected all areas of everyday life including healthcare, as well as in the economic and social domain, and has led to a major decrease in mental health and well-being all over the world [3,4]. As a consequence of these policies, many people have experienced disruptions to their daily routines, including changes in health behaviors such as exercise or dietary habits [5]. Participants in a US-based study, for instance, primarily reported a decrease in physical activity during the early stages of the COVID-19 pandemic [6]. A multi-country study during the first weeks of COVID restrictions revealed that individuals who adapted to restriction measures with a positive change in exercise behavior showed better mental health and well-being than individuals who reported a negative change in exercise behavior, in comparison to before the pandemic [7]. Young adults (18–29 years old) reported more negative changes in exercise behavior, suggesting their health behaviors may have been more negatively impacted by the pandemic.

Furthermore, a general population survey conducted by the Austrian Ministry for Social Affairs, Health, Care, and Consumer Protection during the first lockdown in March 2020 suggested that especially younger people have changed their consumption habits of psychoactive substances during lockdown, with around 40% of 15–34-year-olds having reported an increase in their smoking and drinking behaviors [8]. For adolescents and young people, the pandemic has come at a time of crucial social development and educational and career milestones, and as such they constitute a group that has been especially affected by the pandemic-related measures and restrictions. In Austria, school students were restricted to home schooling, with classes taking place online, from October 2020 to February 2021. Indeed, many studies showed that the mental health of young people has particularly suffered since the emergence of COVID-19 [9].

Some health behaviors, such as exercise, not only improve physical health, but also have protective effects on mental health. For example, a longitudinal study conducted with Chinese college students during the COVID-19 outbreak revealed a mitigating effect of physical activity on mental health [10]. Additionally, a recent cross-sectional study revealed that mental well-being during a COVID-19 lockdown could be improved by maintaining health behaviors such as physical activity and a balanced diet [11]. Adverse health behaviors, however, can negatively affect mental health. For example, alcohol and cigarette consumption as a coping strategy during COVID-19 lockdowns was shown to negatively predict quality of life and well-being, and positively predict perceived stress, depression, anxiety, and insomnia [12]. Therefore, monitoring the health behaviors of young people during the pandemic is important not only for understanding physical health, but also for identifying risk behaviors for mental health. This study aimed to assess the following health behaviors of young people in Austria during the COVID-19 pandemic: smoking, alcohol consumption, cannabis consumption, exercise, and perceived general health status.

Another group that was identified to be at greater risk for decreased mental health during COVID-19 lockdown were women, with more women than men reporting symptoms of depression, anxiety, and insomnia [13]. Moreover, more women than men reported an increase in substance, alcohol, and cigarette consumption during the first lockdown in Austria [8], whereas more men than women reported a decrease in their consumption. However, girls tend to generally exhibit more positive health behaviors than boys [14]. Gender is therefore an important aspect to consider when analyzing health behaviors during the COVID-19 pandemic. In the manuscript at hand, we therefore analyzed if gender had an effect on the health behaviors of Austrian youth. We predicted that health behaviors would differ between genders and explored the assessed health behaviors in each gender separately.

Another hugely important determinant of health behaviors is education level [15]. While the predictive nature of education on positive health behaviors was identified many years ago [16], the reasons for this relationship are not completely clear yet. On the one hand, opportunity costs may affect the relationship between education level and health behaviors [17], on the other hand, education background is also the most important factor for health literacy, which in turn affects health behaviors [18]. In Germany, education was even found to have a causal effect on reducing smoking behavior [19]. We therefore wanted to investigate this in our sample by analyzing whether two different education groups, high school students and apprentices, differed in their health behaviors during the COVID-19 pandemic. We chose to compare these two education groups as apprentices in Austria are typically in the same age group as high school students. Apprenticeship is usually started around the age of 15, after compulsory school has been completed. However, apprenticeship constitutes a different form of dual vocational education, including both vocational school and practical job training, compared to full time high school education. We hypothesized that apprentices would show poorer health behaviors than high school students.

Lastly, vaccinations against COVID-19 are a critical step to ending the pandemic [20]. As health behaviors tend to co-occur [21], it is worth examining the factors that influence adolescents’ willingness to receive a COVID-19 vaccination in order to better inform to whom vaccination campaigns are directed. Results of a recent study, for example, suggest an association between heavy lifetime cannabis use and reduced vaccination willingness; although, no difference in vaccination willingness was found between light and heavy cannabis users [22]. Therefore, we also investigated whether vaccine willingness is affected by other health behaviors in young people and predicted that those with poorer health behaviors would show a lower willingness to receive the COVID-19 vaccine. Overall, we aimed to build a picture of health behaviors in two groups of young people in Austria with differing education types during the COVID-19 pandemic.

## 2. Methods

### 2.1. Study Design

Two cross-sectional online surveys were conducted via REDCap [23]. The first survey concerned apprentices and was run from 29 March to 18 May 2021. A second survey with high school students was carried out from 19 June to 2 July 2021; one semester after home schooling ended and schools reopened again. During the time frame of the first survey, a regional lockdown was in place in the eastern states of Austria (Vienna, Lower Austria, and Burgenland). A 24 h curfew with only few exceptions (coverage of basic needs, work, assistance and care of other people, and outdoor exercise) was reinstated. These measures were lifted on 19 April in Burgenland on 3 May in Vienna and Lower Austria. All other Austrian states set a curfew from 8 p.m. to 6 a.m., containing the same exceptions. Additional measures, such as wearing FFP2-masks in public indoor spaces as well as public transport and keeping a distance of at least two meters from others were in place in all Austrian states. During the second survey, all curfews had been lifted, but mask regulations and social distancing measures remained in place.

Approval for this study was obtained from the ethics committee and the data protection officer of the Danube University Krems (protocol code EK GZ 41/2018-2021). To be included in the study, all participants had to give electronic informed consent by agreeing to the data protection declaration. Furthermore, all participants had to confirm that they were at least 14 years or older.

### 2.2. Measures

The health behaviors were assessed with items based on those used in the HBSC-study [24] to allow for future comparison between the results. As alcohol, tobacco, and cannabis constitute the most commonly consumed psychoactive substances in adolescents [25], they were included in our study.

#### 2.2.1. Exercise

Exercise was assessed by asking the participants on how many of the last seven days they had been physically active for at least 60 min, with possible answers ranging from 0 to 7 days.

#### 2.2.2. Alcohol Consumption

To assess alcohol consumption participants were asked how often they had consumed alcohol during the last 30 days: never, 1–2 days, 3–5 days, 6–9 days, 10–19 days, 20–29 days, daily.

#### 2.2.3. Smoking

Participants were asked how often (if at all) they smoke cigarettes with possible responses; every day, once or more per week but not every day, less than once per week, or I do not smoke.

#### 2.2.4. Cannabis Consumption

Frequency of cannabis consumption during the last 30 days was assessed; daily, once or more per week but not every day, less than once per week, or I do not smoke cannabis.

#### 2.2.5. Perceived Health Status

Health status was measured with the question “How would you describe your health status?” The provided answer options were “excellent”, “good”, “quite good”, and “quite poor”.

#### 2.2.6. Vaccination Readiness

Vaccination willingness was assessed with the single-item question: “If there was the possibility to get vaccinated against the coronavirus, would you do it?”, rated on a 5-point scale from 1 (definitely not) to 5 (definitely).

#### 2.2.7. Other Variables

Age was coded in years. Gender was coded as woman, man or non-binary. Migration background was coded as yes if the participant and/or both parents were born abroad. Region of Austria was coded as state (Vienna, Lower Austria, Upper Austria, Burgenland, Styria, Carinthia, Salzburg, Tyrol, Vorarlberg).

### 2.3. Statistical Analyses

Apprentices and high school students differed significantly in age (t(1721) = 22.49, *p* < 0.0001), gender (χ^2^ = 121.76 *p* < 0.0001), region (χ^2^ = 686.72 *p* < 0.0001), and migration background (χ^2^ = 60.2 *p* < 0.0001). In order to account for these differences between the samples, propensity score (PS) matching was run [26] to balance the differences between the two groups in order for them to be comparable.

The propensity score assigned to each participant represents the probability of belonging to one of the two groups, given a vector of observed covariates [27]. PS matching was conducted in R version 4.1.0 [28] using the MatchIt package [29] and the optimal method, and age, gender, region, and migration background were included as covariates. Subsequent analyses were run on the matched sample.

To assess whether health behaviors differed between apprentices and high school students, general linear models were conducted with smoking, alcohol consumption, cannabis consumption, exercise, and health status as dependent variables and education group (apprentice or high school), gender (woman, man, or non-binary), and propensity score (to account for the matching) as factors, as well as a group*gender interaction. There were too few non-binary participants to analyze in subsequent post-hoc analyses. Following this, a stepwise general linear model was run to assess whether health behaviors influenced the willingness to receive the COVID-19 vaccine: smoking, alcohol consumption, cannabis consumption, exercise, health status, group, and propensity score were included as factors and vaccine readiness was the dependent variable.

## 3. Results

### 3.1. Sample

The study sample consisted of a subsample of 1442 Austrian apprentices (women: 53.5%; men: 45.4%; non-binary: 1.1%; migration background: 29.1%) aged 15 to 43 years (M = 18.19, SD = 2.30), and 563 Austrian high school students (women: 79.6%; men: 18.6; non-binary: 1.6%; migration background: 12.6%) aged 14 to 20 (M = 16.34, SD = 1.33).

All 563 high school students could be matched with apprentices according to age, gender, region, and migration background, resulting in a total sample of 1126. In this sample the high school students were: 79.8% women, 18.8% men, 1.4% non-binary, 12.4% with migration background and a mean age of 16.34 (SD = 1.33). The apprentices were 65.7% women, 33.1% men, 1.2% non-binary, 16.2% with a migration background, and a mean age of 17.23 (SD = 1.47).

### 3.2. Health Behaviors

Table 1 summarizes the health behaviors of high school students and apprentices.

Smoking behavior showed an interaction between group and gender (z(1125) = −2.34, *p* < 0.05). Subset analyses showed that it was significantly affected by group in both women (z(818) = 7.73, *p* < 0.0001) and men (z(291) = 2.87, *p* < 0.01). While 35.7% of apprentices smoke daily compared to 6.6% of high school students, 81.3% of high school students do not smoke, compared to 56.1% of apprentices. This difference was more pronounced in women (Table 1), whereby the difference between high school and apprentice daily smokers was 5.1% versus 37.8%, compared to 12.2% versus 31.2% in men. These values also show that while more male high school students smoke daily than their female colleagues, the opposite was observed for apprentices.

Alcohol consumption also showed an interaction between group and gender (z(1125) = −2.15, *p* < 0.05). Drinking differed by group in women (z(818) = 3.86, *p* < 0.001), but not men (z(291) = 1.72, *p* = 0.08). In apprentices 5.3% reported more than 20 days of drinking in the last month, compared with 2.6% of high school students in the full sample. There was more daily consumption among women in the apprentice group and among men in the high school students (Table 1). Differences between male and female participants were more pronounced in high-school students compared to apprentices. For instance, more female students reported that they did not drink any alcohol within the last month, but more often 1–2 or 3–5 days per week, compared to male students.

There was no group*gender interaction in cannabis use (z(1125) = 1.58, *p* = 0.11). There were slightly fewer regular cannabis consumers among high school students than apprentices, although statistically only a trend (z(1125) = −1.89, *p* = 0.05), and more consumption in men than women (z(1125) = 2.13, *p* < 0.05).

Apprentices did partake in more exercise per week than high school students (t(1122) = −13.31, *p* < 0.0001), with no interaction effect with gender (t(1121) = −0.69, *p* = 0.49), nor a main effect of gender (t(1121) = 1.13, *p* = 0.26). However, high school students rated their health status as better than apprentices (z(1125) = −3.4, *p* < 0.001), again with no interaction with gender (z(1125) = −0.57, *p* = 0.57) nor a main effect of gender (z(1125) = −1.05, *p* = 0.29).

### 3.3. Vaccine Willingness

Of the apprentices, 25.75% stated they would definitely get vaccinated and 23.45% stated definitely not (with the rest stating probably, rather not or don’t know). Twice as many high school students stated they would definitely get vaccinated (55.46%) and only 7.1% said they definitely would not. A general linear model revealed no effect of any health behaviors on willingness to receive the COVID-19 vaccination: smoking χ^2^(3) = 6.82, *p* = 0.08; alcohol χ^2^(6) = 10.0, *p* = 0.12; cannabis χ^2^(6) = 2.69, *p* = 0.85; exercise χ^2^(1) = 0.64, *p* = 0.42; health status χ^2^(3) = 0.71 *p* = 0.87. However, group (apprentice vs. high school) did have an effect when included in the same model, with more high school students than apprentices showing willingness to be vaccinated (χ^2^(1) = 46.66, *p* < 0.0001). See Appendix A (Appendix A) for more details. 

## 4. Discussion

As expected, apprentices and high school students differed significantly in their smoking, alcohol, and exercise behaviors, as well as in perceived health status. Apprentices reported five times higher levels of daily smoking and drinking, and three times higher levels of daily cannabis use than school students. Furthermore, only around 20% of apprentices described their health status as excellent.

These results support our hypothesis that education affects health behaviors and are comparable to those of a survey on the health of Austrian apprentices from 2018/19 [30]. This survey compared the results of apprentices to those of high school students from the HBSC-study [24]. In line with our results, apprentices smoked cigarettes and drank alcohol more often than school students. However, only around 20% of apprentices reported not to have consumed alcohol during the last 30 days in the 2018/19 study, whereas this number has increased to 41% of apprentices in our study. While apprentices and school students from our sample did not significantly differ in their cannabis consumption, there was a trend for more apprentices smoking cannabis, which was a significant effect in the 2018/19 survey, and this difference was greater for girls than for boys. Similarly, in our results, we found the high tobacco smoking levels in apprentices compared to school students to be more pronounced in girls than boys. Furthermore, in our sample more women apprentices also reported daily alcohol consumption, suggesting this group may be particularly at risk for poor health behaviors. In a recent study on the psychosocial consequences of the COVID-19 pandemic on Austrian children and adolescents [31], girls reported higher emotional burden than boys, which may aggravate these differences in health behaviors. On the contrary, cannabis consumption was higher in men than women. Overall, these results are in line with our hypothesis that health behaviors differ by gender, but the precise nature of the influence of gender on health behaviors during the pandemic requires more research.

Apprentices were significantly more physically active than school students in our sample. The 2018/19 survey [30] showed seemingly contradictory results, with more male school students being very physically active than male apprentices, but no differences in moderate exercise between apprentices and school students. However, it is important to note that this survey also differentiated between workplace-related physical activity and recreational exercise, whereas participants in our sample were asked on how many of the last 7 days they had been physically active. Taking this differentiation into account, the 2018/19 study revealed that many apprentices were more physically active due to work related demands, whereas more school students exercised during their free time.

Similar to our results, a French study comparing health behaviors of apprentices and school students in 2008 and 2014 revealed the daily smoking of apprentices to be twice as high as that of school students [32]. Moreover, apprentices in that study were more likely to have used alcohol and cannabis in the last month. Taken together these findings support the argument that education type is an important factor for health behaviors [15], and our results suggest this also holds true in Austria.

Furthermore, as reported by Humer et al. [33], apprentices were less willing to receive a COVID-19 vaccination than school students. Furthermore, women and those with a migration background reported lower willingness to receive the vaccine compared to men and those without migration background, respectively. Our results expanded upon this and when included in the same model as health behaviors, education group still had a significant effect on vaccination willingness in our matched sample, suggesting education to be an important factor to consider when targeting vaccination campaigns. In contrast to our hypotheses, however, none of the included health behaviors significantly affected vaccination willingness. As such, rather than being seen as a health issue per se, the COVID-19 vaccination is seemingly more of a socio-cultural issue.

Some limitations to the study need to be considered when interpreting the results. Although apprentices and high school students were propensity score matched according to the major covariates age, gender, region, and migration background; possible unmeasured confounders may still bias the results. During the time of the first survey concerning apprentices, a regional lockdown was instated, which had been lifted again for the second survey with the high school students. As lockdown measures have been shown to affect adolescent mental health [34,35], this could account for some bias in the results. Furthermore, the cross-sectional nature of the surveys does not allow for causal conclusions about the results.

## 5. Conclusions

In sum, high school students in Austria showed more positive and less negative health behaviors, and additionally perceived their health status as better during the COVID-19 pandemic than apprentices. Education seems to be a protective factor for health behaviors of adolescents and this effect seems to be more pronounced in girls. The results furthermore emphasize the need for interventions aimed at improving health behaviors to support adolescents and especially apprentices during the ongoing COVID-19 pandemic, as it still constitutes a major developmental challenge for adolescents. The changing of restriction measures and policies continue to demand constant adaptation to uncertain circumstances and an unpredictable future. Such adjustment efforts can evoke changes in health behaviors such as physical activity, smoking, and alcohol consumption, which can impact not only physical, but also mental, health. It is therefore crucial to continue monitoring the health behavior patterns of adolescents as the pandemic progresses in order to inform public health and derive appropriate measures to prevent adverse health behavior changes from developing into lasting habits.

## Figures and Tables

**Table 1 ijerph-19-01049-t001:** Percentage of school students and apprentices who engage in each health behavior.

Variable	High School			Apprentice			*p*-Value
	Total	Women	Men	Total	Women	Men	
Smoking	%			%			Gender × group *p* < 0.05 Group difference in women *p* < 0.0001 Group difference in men *p* < 0.01
Never	81.3	82.0	80.2	56.1	54.9	59.1	
Less than 1× per week	8.7	9.1	6.6	4.1	4.6	3.2	
1× or more per week	3.4	3.8	0.9	4.1	2.7	6.5	
Daily	6.6	5.1	12.3	35.7	37.8	31.2	
Alcohol—last 30 days	%			%			Gender × group *p* < 0.05 Group difference in women *p* < 0.001 Group difference in men *p* = 0.08
None	30.9	29.4	35.8	40.9	41.4	40.3	
1–2 days	27.9	29.8	20.8	22.9	22.7	23.1	
3–5 days	22.6	24.3	17.0	16.2	16.5	16.1	
6–9 days	10.8	10.0	14.2	8.7	9.2	8.1	
10–19 days	5.2	4.5	6.6	6.0	5.4	7.0	
20–29 days	2.1	1.8	3.8	3.0	2.4	4.3	
30 days	0.5	0.2	1.9	2.3	2.4	1.1	
Cannabis—last 30 days	%			%			Gender × group *p* = 0.11 Main effect group *p* = 0.05 Main effect gender *p* < 0.05
None	92.2	93.8	86.8	86.1	86.8	85.5	
1–2 days	3.9	3.6	5.7	5.2	5.1	5.4	
3–5 days	1.1	0.7	1.9	1.4	1.1	1.6	
6–9 days	0.5	0.7	0	0.5	0.5	0.5	
10–19 days	1.2	0.7	2.8	2.0	1.9	2.2	
20–29 days	0.4	0.2	0.9	2.8	3.0	2.2	
30 days	0.7	0.4	1.9	2.0	1.6	2.7	
Exercise—mean days per week	1.63	1.63	1.68	3.24	3.15	3.46	Gender × group *p* = 0.49 Main effect group *p* < 0.0001 Main effect gender *p* = 0.26
Health status	%			%			Gender × group *p* = 0.57 Main effect group *p* < 0.001 Main effect gender *p* = 0.29
Excellent	32.5	30.7	41.5	22.6	21.6	25.3	
Good	40.3	39.0	44.3	45.6	46.5	44.6	
Quite good	19.4	21.6	10.4	24.7	25.1	24.2	
Bad	7.8	8.7	3.8	7.1	6.8	5.9	

## Data Availability

All relevant data are provided upon request (teresa.orourke@donau-uni.ac.at).

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
