# Peer review of "Health Behaviors in Austrian Apprentices and School Students during the COVID-19 Pandemic"

_ijerph, 2022, doi:10.3390/ijerph19031049_

Round 1
Reviewer 1 Report
This is a study that aimed to assess health behaviors in Austrian high school students and learners during the COVID-19 pandemic. And it also explored whether the willingness to get vaccinated is affected by these behaviors.
Authors must further elaborate the reason why they decided to compare apprentices with high school students, since statistical procedures even had to be used to make the groups comparable. It is also not specified whether these groups of students had been taking classes online or not, as happened in other countries.
Line 114 uses the APA referencing style and not the Vancouver style, and this citation is not in the reference list.
The presentation of results should be improved. Thus, p values ​​ should be included in the tables to make the results clearer. Before presenting the prediction model of the intention to vaccinate, the results for both groups in percentages must be shown.
Author Response
Dear Reviewer, thank you for your feedback. We hope we have addressed your comments to your satisfaction. Please see here our responses.
This is a study that aimed to assess health behaviors in Austrian high school students and learners during the COVID-19 pandemic. And it also explored whether the willingness to get vaccinated is affected by these behaviors.
Authors must further elaborate the reason why they decided to compare apprentices with high school students, since statistical procedures even had to be used to make the groups comparable.
- Thank you for this suggestion. We have added more information on our choice to compare high school students and apprentices to the manuscript (line 96). “We chose to compare these two education groups as apprentices in Austria are typically in the same age group as high school students. Apprenticeship is usually started around the age of 15, after compulsory school has been completed. However, apprenticeship constitutes a different form of dual vocational education, including both vocational school and practical job training, compared to full time high school education.”
It is also not specified whether these groups of students had been taking classes online or not, as happened in other countries.
- Thank you for these relevant suggestions. We agree with the reviewer and hope we have resolved any ambiguity by adding more information to the introduction, line 61 (“In Austria, school students were restricted to home schooling, with classes taking place online from October 2020 to February 2021.”) and the study design section, line 117 (“A second survey with high school students was carried out from June 19th to July 2nd, 2021, one semester after home schooling ended and schools reopened again.”).
Line 114 uses the APA referencing style and not the Vancouver style, and this citation is not in the reference list.
- Thank you for pointing out this mistake! We have now changed this citation to the Vancouver style to match the rest of the manuscript.
The presentation of results should be improved. Thus, p values should be included in the tables to make the results clearer. Before presenting the prediction model of the intention to vaccinate, the results for both groups in percentages must be shown.
- The p-values from the statistical tests have now been added to Table 1.
- We agree the percentages for vaccination willingness should have been included and we have now added this to the results as suggested (line 240). A table has also been added to the supplementary materials with a complete breakdown of vaccine willingness according to health behaviours and education group.
Reviewer 2 Report
Although it is a work that provides relevant information to the scientific community, I consider that some aspects should be taken into account:
Introduction: What is the difference between learners and students, it is not clear to the readers of the article.
Why you are only asked about cannabis and tobacco?
I suggest a very brief reflection on previous studies.
Need reference: The COVID-19 outbreak in December 2019 and the following restrictions have had detrimental effects on the lives of people all over the globe
Need reference: These measures have rapidly affected all areas of everyday life, including healthcare as well as the economic and social domain and led to a major decrease in mental health and well-being all over the world.
Author Response
Dear Reviewer, thank you for your comments and suggestions. We hope we have addressed all of your points to your satisfaction. Please see here our responses.
Although it is a work that provides relevant information to the scientific community, I consider that some aspects should be taken into account:
Introduction: What is the difference between learners and students, it is not clear to the readers of the article.
- Thank you for pointing out this missing information. We have added more information to the manuscript (line 96) and hope we have cleared up this aspect for readers of the article. “We chose to compare these two education groups as apprentices in Austria are typically in the same age group as high school students. Apprenticeship is usually started around the age of 15, after compulsory school has been completed. However, apprenticeship constitutes a different form of dual vocational education, including both vocational school and practical job training, compared to full time high school education.”
Why you are only asked about cannabis and tobacco?
- Thank you for bringing up this question. As mentioned in the measures section, line 134, „We assessed health behaviors with items based on those used in the HBSC-study, to allow for future comparison between the results.“. For this age group alcohol, cigarette and cannabis constitute the most commonly used psychoactive substances (e.g. Strizek et al., 2008), which is why we included them in our survey. To clarify this further, we have added this information to the methods section, line 135.
I suggest a very brief reflection on previous studies.
- Thank you for this suggestion. We agree on the importance of reflecting previous studies. We have covered health behaviors and education level in that age group and have added references as suggested.
Need reference: The COVID-19 outbreak in December 2019 and the following restrictions have had detrimental effects on the lives of people all over the globe
- Thank you for pointing this out. We agree and have added the following reference to support this statement:
Chakraborty, I.; Maity, P. COVID-19 outbreak: Migration, effects on society, global environment and prevention. Sci. Total Environ. 2020, 728, 138882
Need reference: These measures have rapidly affected1 all areas of everyday life, including healthcare as well as the economic and social domain and led to a major decrease in mental health and well-being all over the world.
- We agree with the reviewer and have added the following references to support this statement:
Brooks, S.K.; Webster, R.K.; Smith, L.E.; Woodland, L.; Wessely, S.; Greenberg, N.; Rubin, G.J. The psychological impact of quarantine and how to reduce it: Rapid review of the evidence. Lancet 2020, 395, 912-920.
Galea, S.; Merchant, R.M.; Lurie, N. The Mental Health Consequences of COVID-19 and Physical Distancing: The Need for Prevention and Early Intervention. JAMA Intern. Med. 2020, 180. 817-818)
Reviewer 3 Report
Thanks for inviting me to review this paper that addresses differences in the health behaviors between apprentices and school students during the COVID-19 pandemic in Austria. The authors also tended to analyze the correlation between vaccination willingness and health behaviors. I found the topic interesting however the findings were difficult to follow, largely because the methods were not explicitly laid out in the paper. Comments are followed.
- Please describe explicitly the regression analyses on the association between vaccination willingness and predictors. Were these analyses stepwise, hierarchical, or multilevel?
- There seems a missing table of the regression results, although the results may be not significant on many predictors according to the texts. A table can, however, tell more details.
- In the discussion, please comment on why apprentices have a tendency to forego vaccination.
Author Response
Dear Reviewer, thank you for your comments and suggestions. We hope we have addressed all of your points to your satisfaction. Please see here our responses.
Thanks for inviting me to review this paper that addresses differences in the health behaviors between apprentices and school students during the COVID-19 pandemic in Austria. The authors also tended to analyze the correlation between vaccination willingness and health behaviors. I found the topic interesting however the findings were difficult to follow, largely because the methods were not explicitly laid out in the paper. Comments are followed.
- Please describe explicitly the regression analyses on the association between vaccination willingness and predictors. Were these analyses stepwise, hierarchical, or multilevel?
- Thank you for encouraging a more detailed approach to statistical reporting. The analyses were stepwise and this information has now been added to the statistical analyses section of the methods (line 191).
- There seems a missing table of the regression results, although the results may be not significant on many predictors according to the texts. A table can, however, tell more details.
- This table has now been added as a supplementary file.
- In the discussion, please comment on why apprentices have a tendency to forego vaccination.
- We agree that this is an important aspect and have commented on this relationship in line 291: “Furthermore, as reported by Humer et al. [33], apprentices were less willing to get a COVID-19 vaccination than school students. Furthermore, women and those with a migration background reported lower willingness to receive the vaccine compared to men and those without migration background respectively. Our results expanded upon this and when included in the same model as health behaviors, education group still had a significant effect on vaccination willingness in our matched sample, suggesting education to be an important factor to consider when targeting vaccination campaigns. “ We therefore think that education has more of an impact than health behaviors in apprentices tendency to forego vaccination.
- As this was not the primary focus of our paper, we did not explore this relationship any further, but recommend investigating the complex associations between education and vaccination willingness in future studies.
Round 2
Reviewer 1 Report
No comments. The authors have taken into account all of my suggestions
Reviewer 3 Report
Thanks for the effort to address my concerns. I have no more questions.